# Cell-Free DNA Extracted from CSF for the Molecular Diagnosis of Pediatric Embryonal Brain Tumors

**DOI:** 10.3390/cancers15133532

**Published:** 2023-07-07

**Authors:** Mathieu Chicard, Yasmine Iddir, Julien Masliah Planchon, Valérie Combaret, Valéry Attignon, Alexandra Saint-Charles, Didier Frappaz, Cécile Faure-Conter, Kévin Beccaria, Pascale Varlet, Birgit Geoerger, Sylvain Baulande, Gaelle Pierron, Yassine Bouchoucha, François Doz, Olivier Delattre, Joshua J. Waterfall, Franck Bourdeaut, Gudrun Schleiermacher

**Affiliations:** 1Recherche Translationelle en Oncologie Pédiatrique (RTOP), INSERM U830 Cancer, Heterogeneity, Instability and Plasticity, Department of Translational Research, Institut Curie Research Center, PSL Research University, 75005 Paris, France; 2Unité de Génétique Somatique, Service de Génétique, Institut Curie Hospital Group, 75005 Paris, France; 3Plateforme de Génomique des Cancers, Centre Léon Bérard, 69008 Lyon, France; 4Laboratoire de Recherche Translationnelle, Centre Léon-Bérard, 69373 Lyon, France; 5Department of Pediatric Clinical Trials and Department of Pediatric Neuro-Oncology, Institut d’Hématologie et d’Oncologie Pédiatrique, 69008 Lyon, France; 6Department of Pediatric Neurosurgery, Hôpital Necker-Enfants Malades, Assistance Publique Hôpitaux de Paris-Université Paris Cité, 75015 Paris, France; 7GHU Psychiatrie et Neurosciences, Site Sainte-Anne, 75014 Paris, France; 8Department of Pediatric and Adolescent Oncology, Gustave Roussy Cancer Campus, Université Paris-Saclay, 94805 Villejuif, France; 9Institut Curie Genomics of Excellence (ICGex) Platform, Institut Curie Research Center, 75005 Paris, France; 10SIREDO Integrated Pediatric Oncology Center, Institut Curie Hospital Group, 75005 Paris, France; 11Faculty of Medicine, Université Paris Cité, 75005 Paris, France; 12Diversity and Plasticity of Childhood Tumors Laboratory, INSERM U830 Cancer, Heterogeneity, Instability and Plasticity, Institut Curie Research Center, PSL Research University, 75005 Paris, France; 13Integrative Functional Genomics of Cancer Laboratory, INSERM U830 Cancer, Heterogeneity, Instability and Plasticity, PSL Research University, 75005 Paris, France; 14Department of Translational Research, Institut Curie Research Center, PSL Research University, 75005 Paris, France

**Keywords:** pediatric embryonal brain tumors, liquid biopsy, cell-free DNA, molecular diagnosis, nucleosome footprinting

## Abstract

**Simple Summary:**

We demonstrate that, in pediatric embryonal brain tumors, cell-free DNA extracted from CSF can be used for whole exome sequencing (WES), with informative results in 83% of samples. Importantly, in comparison to the WES of primary tumor tissue, clonal heterogeneity is identified in most cases. In a novel approach, nucleosome footprinting at transcription start sites of genes of interest enables the inference of gene expression. These results pave the way for the use of CSF cfDNA for molecular diagnosis and disease monitoring.

**Abstract:**

Background: Liquid biopsies are revolutionary tools used to detect tumor-specific genetic alterations in body fluids, including the use of cell-free DNA (cfDNA) for molecular diagnosis in cancer patients. In brain tumors, cerebrospinal fluid (CSF) cfDNA might be more informative than plasma cfDNA. Here, we assess the use of CSF cfDNA in pediatric embryonal brain tumors (EBT) for molecular diagnosis. Methods: The CSF cfDNA of pediatric patients with medulloblastoma (*n* = 18), ATRT (*n* = 3), ETMR (*n* = 1), CNS NB FOXR2 (*n* = 2) and pediatric EBT NOS (*n* = 1) (mean cfDNA concentration 48 ng/mL; range 4–442 ng/mL) and matched tumor genomic DNA were sequenced by WES and/or a targeted sequencing approach to determine single-nucleotide variations (SNVs) and copy number alterations (CNA). A specific capture covering transcription start sites (TSS) of genes of interest was also used for nucleosome footprinting in CSF cfDNA. Results: 15/25 CSF cfDNA samples yielded informative results, with informative CNA and SNVs in 11 and 15 cases, respectively. For cases with paired tumor and CSF cfDNA WES (*n* = 15), a mean of 83 (range 1–160) shared SNVs were observed, including SNVs in classical medulloblastoma genes such as SMO and KMT2D. Interestingly, tumor-specific SNVs (mean 18; range 1–62) or CSF-specific SNVs (mean 5; range 0–25) were also observed, suggesting clonal heterogeneity. The TSS panel resulted in differential coverage profiles across all 112 studied genes in 7 cases, indicating distinct promoter accessibility. Conclusion: CSF cfDNA sequencing yielded informative results in 60% (15/25) of all cases, with informative results in 83% (15/18) of all cases analyzed by WES. These results pave the way for the implementation of these novel approaches for molecular diagnosis and minimal residual disease monitoring.

## 1. Introduction

Pediatric central nervous system (CNS) tumors are the most common solid tumors in children and comprise 15% to 20% of all malignancies in children. Among these, embryonal brain tumors (EBT) represent a complex and highly aggressive group, including entities such as medulloblastoma (MB), atypical teratoid rhabdoid tumors (ATRT) and embryonal tumors with multilayered rosettes (ETMR), as well as recently described embryonal tumors with *FOXR2* activation or *BCOR* alteration [1,2,3].

MB, the most prevalent EBT, constitutes a heterogeneous disease with four main groups associated with variable outcomes [4]. The combination of histological and molecular features is now systematically used in the clinical management of patients with MB. WNT MB are mostly characterized by *CTNNB1* or *APC* mutations and the monosomy of chromosome 6; Sonic hedgehog (SHH) MB can harbor genomic alterations in several genes including *PTCH1*, *SUFU*, *MYCN*, *ELP1* and *TP53*; group 3 and group 4 MBs have few specific genomic alterations, but *MYC* amplification in group 3 MB is an important prognostic marker associated with a poor outcome [5].

ATRT is the secondmost frequent EBT; this aggressive disease occurs in early childhood either in infra- or supra-tentorial regions. The genomic hallmark of ATRT is the biallelic inactivation of *SMARCB1* with an otherwise extremely stable genome, whereas three epigenetically/transcriptionally defined subtypes have been identified [6]. The SHH ATRT subgroup is characterized by an overexpression of *GLI2*, TYR ATRT is characterized by an overexpression of several melanosomal markers and *OTX2* and MYC ATRT is characterized by an overexpression of the *MYC* oncogene.

The very aggressive ETMR are supra- or infra-tentorial tumors of early childhood and share features with the also aggressive CNS BCOR-ITD; while ETMR are characterized by the amplification of an miRNA cluster on chromosome 19 (C19MC amplification), CNS BCOR-ITD harbor an internal tandem duplication of the *BCOR* gene. CNS Neuroblastoma FOXR2-activated tumors show complex rearrangements of the *FOXR2* locus.

Finally, other rare EBT, defined by their specific histopathological features together with their genomic, expression-based or methylation-based profiling, have also been described. Other rare EBT, NOS (not otherwise specified), are defined by the absence of diagnostic criteria qualifying histologically and molecularly defined CNS EBT [4,7].

Tumor characterization based on pathological and molecular analyses of primary tumor tissue obtained at diagnosis by either frontline surgery or biopsy is of high importance in determining the exact diagnosis and risk group. Indeed, several of these genetic alterations such as *SMARCB1* inactivation, *BCOR* ITD or the amplification of C19MC are considered specific enough to allow for a definitive diagnosis [3]. Multi-omic profiling might also contribute to the identification of predictive biomarkers, the introduction of novel targeted therapies at relapse and, importantly, upfront novel treatment approaches [8,9].

However, a lack of sufficient tumor samples, low-quality nucleic acid content following formalin fixation or the presence of non-tumor tissue can create inaccuracies in tumor classification and hamper detailed molecular explorations. Occasionally, the neurosurgical procedures (resection or biopsy) are deemed too risky for the patient, and treatment may start without a definitive documented diagnosis based on tissue sampling. Therefore, there is an ongoing need to improve the approaches for the characterization of EBT at diagnosis and to document further molecular alterations in case of relapse.

Liquid biopsies represent recent, noninvasive methods for characterizing tumors. The study of cell-free DNA (cfDNA), small DNA fragments released into either the blood stream or other body fluids from cells undergoing apoptosis, necroptosis or other cellular degradation processes, has led to the recent development of revolutionary tools for molecular diagnosis. The study of circulating tumor DNA (ctDNA), a variable fraction of the overall cfDNA, extracted from blood plasma is rapidly moving into standard clinical care for cancer patients. However, for patients with brain tumors, varying amounts of ctDNA have been demonstrated in cfDNA extracted from plasma, and it is thought that the blood–brain barrier might limit the detection of tumor-specific DNA fragments in the blood [10]. Importantly, recent reports have indicated that significant amounts of cfDNA can be extracted from cerebrospinal fluid (CSF) and that significant amounts of ctDNA can be detected in patients with different types of brain tumors, including pediatric EBT [2,11,12,13,14]. However, to date, only limited data have been reported on the use of cfDNA extracted from CSF for the complete molecular characterization of pediatric EBT [12,13,15].

In addition to genetic analysis, cfDNA can also enable the inference of epigenetic features. Methylation profiles have been generated from the CSF cfDNA of brain tumor patients [16]. Recent approaches have explored the feasibility of the study of nucleosome footprints in cfDNA based on the coverage in whole-genome sequencing (WGS) data around nucleosome positions. It is hypothesized that a dense occupation by nucleosomes, corresponding to a silencing of expression at transcription start sites (TSS), might protect DNA from degradation by nucleases and that the resulting coverage patterns after sequencing might identify the tissue of origin and the prediction of expression patterns [17,18,19,20].

Altogether, liquid biopsies are revolutionary tools for detecting tumor-specific genetic and epigenetic alterations in body fluids. Here, we assess whether the cfDNA in CSF could be used for the molecular diagnosis of pediatric EBT.

## 2. Patients and Methods

### 2.1. Patients

Patients with embryonal brain tumors treated at Institut Curie or Centre Leon Berard (CLB) were included in this study if CSF was available for the extraction of cfDNA (Appendix A). Twenty-five cases were identified. The CSF samples were obtained at diagnosis for all patients except one, for whom the sample was obtained at relapse. Treatment was given according to national or international protocols according to the disease type. For all patients, written informed consent was obtained from parents/guardians according to national law. In addition to clinical molecular analyses, comprehensive molecular characterization was performed following inclusion in the national MICCHADO (NCT03496402) or MAPPYACTS studies (NCT02613962) [8] in two and three cases, respectively. This study was approved by the Institutional Review Board of Institut Curie (Reference DATA210043).

### 2.2. Sample Collection and Processing

For each patient, genomic DNA was extracted from either a surgical or a biopsy tumor sample obtained during surgery, or biopsy, at diagnosis or disease recurrence according to standard procedures. CSF was obtained during a clinically indicated lumbar puncture, with a mean delay between surgery and lumbar puncture of 17.9 days (range: 3 days before to 67 days after surgery/biopsy). In one case (patient 8), CSF was obtained after the first two cycles of chemotherapy. Cytological analysis revealed the presence of tumor cells in the CSF in 5/25 samples. For cfDNA studies, a minimum of 300 µL of CSF was prepared by centrifugation at 2000 rpm for 10 min, followed by careful aliquoting of the supernatant and freezing at −80 °C within 1 to 24 h after collection. Germline DNA extracted from blood leucocytes was available in six cases [21].

### 2.3. Molecular Analysis of Primary Tumors

The clinical molecular diagnosis of primary pediatric EBT involved array-CGH, targeted sequencing and/or Nanostring profiling, according to the tumor entity, as previously reported [22,23,24]. Whole-exome sequencing (WES) of genomic DNA extracted from primary tumors, and paired germline genomic DNA, was performed in a total of 18 and 6 patients, respectively (Appendix A) [8].

### 2.4. cfDNA Extraction, Library Construction and Whole-Exome Sequencing of cfDNA

cfDNA was extracted from a minimum of 300 µL of CSF using a QIAamp Circulating Nucleic Acid Kit (Qiagen, Courtaboeuf, France) [25]. The total cfDNA concentration per mL of CSF was calculated. cfDNA sequencing libraries were constructed without fragmentation. The Medexome Enrichment Kit (Nimblegen™, Roche Diagnostics, Meylan, France) was used for whole-exome capture. Paired-end (100 bp) Illumina™ WES (Illumina, Paris, France) was performed (expected coverage of 100×; Appendix A).

### 2.5. Targeted Sequencing Panel Design

For deep coverage targeted sequencing for inferring expression based on the coverage around the TSS, a custom panel was designed to cover EBT-relevant genes and their TSS. A total of 112 genes that contribute to the genetic classification of embryonic tumors such as *MYC* (in the MB and ATRT MYC subtype), *MYCN* (in the MB and ATRT SHH subtype), *TYR* (in the ATRT TYR subtype) or *JAK3* were used for a diagnostic Nanostring panel design at Institut Curie [7,26,27]. The regions (+/−1 kb) surrounding the TSS of these genes were included in the design. In addition, the coding sequences of six genes (*CTNNB1*, *PTCH1*, *SMARCB1*, *SMARCA4*, *SMO* and *SUFU*) were added, as well as five glioma hotspots mutations (*H3F3A*_K27M, *HIST1H3B*_K27M, *BRAF*_V600E, *IDH1*_R132C and *IDH2*_R172K) [28,29]. For the prediction of expression, eight TSS were selected as controls: (1) four ubiquitously expressed genes with strong or moderate expression (*ACTB*, *B2M*, *GAPDH* or *SDHA431)*, (2) two genes not expressed in the central nervous system (*BAGE4* and *ACPT*) and (3) two genes expressed in the CNS (*SLC32A1* and *CNTNAP2*) were selected from the Protein Atlas database [30,31]. This panel design, referred to as the targeted sequencing TSS panel, was used on CSF samples. Sequencing was performed to a targeted coverage of 500×.

### 2.6. Bioinformatics Analysis

#### 2.6.1. Copy Number and SNV Analysis

The WES and target sequencing raw reads were mapped to the reference human genome assembly GRCh37/hg19 using BWA v0.7.15, with default parameters, and coverage scores were computed [32]. Variants were called with GATK 4.1.7.0/Mutect2 and annotated, and further filters were applied prior to IGV visual inspection (see Appendix A for details). Copy number analyses were performed with snp-pileup_0.5.14 and FACETS v0.5.11.

#### 2.6.2. Sequencing across the TSS to Infer Expression Profiles

Bigwig files containing the coverage scores representing the number of reads per nucleotide were generated with the option BAM coverage using BAM files [32].

Matrices with a mean score per genomic region were computed. The capture regions of the TSSs had a size of 2 kb (−1 kb downstream and +1 kb upstream of the TSS) to include the nucleosome depleted region (NDR). K-means clustering using deeptools, with the number of clusters set to two, was used to classify genes based on the coverage across the TSS [18,19].

#### 2.6.3. Sequencing Coverage, Quality Statistics and Data Availability

For WES and targeted sequencing panel analysis, the sequencing coverage and quality statistics for each sample are summarized in Appendix A, respectively. The reference genome assembly GRCh37 was used to map the reads. Sequencing data (.vcf files) will be made available upon reasonable request.

## 3. Results

Tumor samples and CSF from 25 pediatric EBT patients were analyzed in this study: medulloblastoma (*n* = 18), ATRT (*n* = 3), ETMR (*n* = 1), CNS NB FOXR2 (*n* = 2) and pediatric EBT NOS (*n* = 1) (Appendix A).

### 3.1. cfDNA Concentrations

Following cfDNA extraction from CSF, variable cfDNA concentrations were obtained (mean 48 ng/mL of CSF; range 4–442 ng/mL), independent of the tumor type (Appendix A). No correlation between the cfDNA concentration and the age, disease status (localized versus metastatic disease), presence/absence of a postsurgical residue or delay between surgery and CSF sampling was observed (Appendix A). The extracted cfDNA was then used for molecular characterization by different sequencing approaches.

### 3.2. Informative Sequencing Approaches and Determining the ctDNA Fraction in CSF cfDNA

Primary tumor DNA was sequenced by targeted sequencing using an in-house diagnostic panel in 7 cases (combined with aCGH), WES in 17 cases and both approaches in 1 case, with paired germline available for WES in 6 cases. These approaches identified large-scale CNAs in tumor DNA in all cases except one and SNVs in all cases (Figure 1 and Appendix A). Thus, primary tumor analysis was informative in all cases. For CSF cfDNA analysis, the sequencing results were considered informative in the case of the identification of at least one CNA or SNV. cfDNA extracted from CSF was sequenced by WES alone in ten cases, the TSS targeted panel alone in seven cases or both approaches in eight cases. These approaches identified CNAs and SNVs in 11 cases and SNVs only in 4 additional cases (Figure 1, Appendix A). No correlation between the cfDNA concentration and the total number of CNAs and SNVs was observed (Appendix A). In 10 CSF cfDNA samples (the 7 samples analyzed by targeted sequencing alone and 3 samples sequenced by WES, with very low coverage), no CNA nor SNV could be detected.

The fraction of ctDNA in the CSF cfDNA was determined in samples sequenced by WES, with a range of 0.15–0.86 (Appendix A). The highest ctDNA content was observed in the ETMR case.

### 3.3. Molecular Characterization: Copy Number Profiling

In the 11 cases with informative copy number profiles in both the primary tumor and CSF cfDNA samples, the comparison of CNA between the primary tumor and the cfDNA showed an excellent concordance (Figure 2), with amplifications of *MYCN*, a region on chromosome 20 and a region on chromosome 19 (*n* = 4), homozygous deletions or heterozygous copy number losses or gains, including the loss and gain of chromosome 17p. CNAs detected only in the primary tumor were seen in four cases and involved heterozygous copy number changes (loss of chromosome 6, deletions of chromosome 1p, 10q, 16q, gain of chromosome 7, 8q, 10p).

### 3.4. Molecular Characterization: SNVs

In 15 cases with informative SNV analysis in the primary tumor and CSF cfDNA, a comparison of the variant allele frequency (VAF) of SNVs showed a good correlation (r = 0.5, *p* < 0.0001; Figure 3). However, in some cases, SNVs were seen only in the tumor and not in the corresponding CSF cfDNA, including SNVs in *ROBO1*, *APC*, *PIK3CA* or *ARID1A*. In other cases, SNVs in genes of interest were detected only in the CSF cfDNA but not in the primary tumor, including SNVs in the genes *BCORL1* or *NCOR1*.

### 3.5. Diagnostic Genetic Alterations Can Be Identified in CSF cfDNA

Next, we sought to determine whether molecular genetic alterations routinely used in diagnosis or subgrouping could also be detected in the CSF cfDNA. In 3/3 WNT MBs, we identified the typical subgroup driver mutations, i.e., two *CTNNB1* heterozygous mutations (c.100G>A/p.(Gly34Arg) and c.98C>G/p.(Ser33Cys)) and one *APC* homozygous mutation (c.3758del/p.(Ser1253Leufs*12)). In 2/3 ATRT, the *SMARCB1* mutation and/or deletion was found in CSF cfDNA. However, in the 1/3 ATRT case, CSF cfDNA analysis did not detect the previously identified *SMARCB1* mutation, and in 1 MB, 1 NB-FOXR2 and 1 EBT NOS, CSF cfDNA analysis did not enable the detection of typical *PTCH1*, *PTPRK* and *TP53/KRAS* mutations, respectively (Table 1). The typically diagnostic C19MC amplification was identified from CSF cfDNA in 1/1 ETMR and the suggestive 1q gain in 1/2 NB-FOXR2. Altogether, in 11/25 cases, previously identified diagnostic SNV or CNA could be identified in CSF cfDNA.

### 3.6. Intratumor Heterogeneity

Taking into account all SNVs observed among the 15 cases with informative SNV analysis in the primary tumor and CSF cfDNA by WES, a mean of 83 (range 1–60) SNVs were observed in common between the tumor and CSF cfDNA. Furthermore, a mean of 5 SNVs (range 0–25) and 18 SNVs (range 1–62) were seen only in the CSF cfDNA or only in the tumor, respectively, indicating clonal heterogeneity in all EBT types (Figure 4). Importantly, copy number analyses also indicated heterogeneity, with an *MYC* amplification seen in the CSF cfDNA but not the primary tumor, later emerging at the time of relapse (patient 17; Appendix A).

### 3.7. Inferring Expression Patterns for Tumor Classification Based on the Coverage around the tSS

To study the nucleosome footprints of TSS in the cfDNA samples, the targeted sequencing TSS panel was used for sequencing CSF cfDNA in 15 cases. The observed mean coverage across the targeted sites was 194× (range 4–628). Interestingly, different profiles of coverage across the TSS were observed, ranging from a uniform coverage across the entire captured region to a dip in coverage over the nucleosome-depleted region (NDR).

For seven samples, the coverage profile clearly distinguished two different patterns, suggesting two distinct states of nucleosome occupancy (Figure 5A). In these seven samples, where the concentration of cfDNA was >10 ng/mL (Figure 5C), TSS clustering was deemed informative for downstream analysis.

In eight other samples, these two distinct patterns could not be clearly distinguished (Figure 5B). Seven of these samples had a library size below 1 million reads. For all eight samples, the concentration of cfDNA was <10 ng/mL, suggesting that the concentration of cfDNA highly impacts the TSS footprint analysis and clustering (Figure 5C). These eight cases were deemed not informative for TSS clustering.

We then performed a global heatmap on the NDR regions of the 112 genes for which we captured the TSS. This enabled hierarchical clustering of the coverage patterns. The heatmap shows differences in coverage around the TSSs across the seven samples (Figure 5D), reflecting differences in nucleosome occupancy and identifying genes that might be “expressed” as opposed to others that might be “silent”. Across the seven informative samples, nearly all genes had variations in the TSS coverage profiles, with an identical coverage only observed for one gene.

The TSS clustering of the different samples, taking into account the disease type, showed that two out of three MB samples clustered together, whereas other disease types were more dispersed (Appendix A).

For downstream analysis, we focused on the set of genes for which the coverage scores were consistently close between ATRT cases and different from MB cases and *v*/*v*, to determine a subset of genes for which TSS nucleosome occupancy might be “ATRT-specific” and “MB-specific”. To achieve this, the normalized coverage scores (row z-score), identifying a threshold that allows for the classification of the genes into “expressed” and “silent”, were determined. With scores ranging from −1.83 to 2.24, values under 0 are considered expressed values, and those over 0 are considered silent. This definition of TSS coverage specific to ATRT or MB identified 46 genes with a clear difference between the disease types ATRT and MB (Figure 5E). Six of these genes show an NDR coverage profile suggesting gene expression in MB but not in ATRT (*PDLIM3*, *RELL1*, *EMX2*, *UNC5D*, *FOXN3*, *BAGE3*), with 40 others showing an NDR coverage suggesting gene expression in ATRT but not in MB. However, no correlation with expression data determined by Nanostring was observed.

Altogether, somatic genetic alterations (SNVs and/or CNAs) could be observed in 15/25 CSF cfDNA samples, and specific nucleosome footprints were observed in 7/15 cases.

## 4. Discussion

Molecular characterization is an integral part of the diagnostic procedures of CNS EBT [3,4]. Nevertheless, in many instances, the quantity of available tumor tissue is limited. Furthermore, modifications in genetic and epigenetic profiles might occur under therapy, with limited possibilities of exploration.

Liquid biopsies are revolutionary tools for detecting tumor-specific genetic alterations in body fluids, and cfDNA can be used for molecular diagnosis, follow-ups of disease burden or the study of clonal evolution across many different solid malignancies, including pediatric solid tumors such as neuroblastoma and rhabdomyosarcoma [8,25]. In EBT, to date, only a few studies have addressed the feasibility and clinical utility of CSF cfDNA [15,33,34,35]. In particular, low-coverage WGS has been used for the calling of tumor-specific CNAs at diagnosis and as a surrogate of minimal residual disease in MB [15,33].

We confirm, in our series, that CSF can be a reliable source of cfDNA in pediatric EBT, although obtained in small quantities. Furthermore, we show that this cfDNA, used for WES in 18/25 cases, leads to informative results for CNA and SNV identification in 11/25 and 15/25 cases, respectively. Non-informative cases might be linked to the small amounts of CSF, the low ctDNA content or the delay until CSF collection after surgery, particularly for cases with complete excision (mean of 17.9 days after surgery; range −3 days to +67 days). In prospective studies, CSF collection at diagnosis, either before or during surgery, might increase the utility of cfDNA analysis.

In our series, CNAs could be detected in 11/18 CSF cfDNA samples for which the performed techniques permitted the calling of copy number changes [34]. All cases with a ctDNA content > 20% showed copy number changes in the CSF cfDNA. Furthermore, SNVs could be detected in 15/18 cases for which WES was performed, confirming the higher sensitivity for the detection of SNVs as compared to CNA.

Our results also demonstrate genetic clonal heterogeneity, with tumor- and CSF-specific SNVs observed in all EBT, underlining the usefulness of CSF cfDNA for further evaluating this phenomenon [5,36].

Given the feasibility of the molecular characterization of EBT using CSF cfDNA, these approaches might be of clinical usefulness in addressing multiple questions in the future. First, in cases where a tissue biopsy is impossible or deemed too risky, analyzing cfDNA could lead to a clinical diagnosis and may be of particular use for distinguishing between a tumor relapse and a secondary malignancy. Second, these approaches are feasible for monitoring tumor burdens and the evaluation of minimal residual disease [33,37]. Furthermore, the study of CSF cfDNA paves the way towards the longitudinal evaluation of genetic alterations, as temporal heterogeneity has been described in EBT [36]. In addition, this will enable the study of the emergence of resistance mutations or other genetic alterations under (targeted) therapies [25]. Further studies might also explore the possibility of the analysis of cfDNA extracted from blood in pediatric patients with EBT.

In addition to molecular diagnosis based on CNA or SNVs, the classification of EBT strongly depends on methylation and expression profiles [38,39,40]. Due to the limited amount of cfDNA available for each sample, the methylation profiling on CSF cfDNA could not be performed. We sought to determine whether, in CSF, cfDNA expression profiles could be inferred from nucleosome footprints based on the sequencing coverage across the TSS of genes of interest. Nucleosome occupancy at transcription start sites (TSS) can be reflected by cutting patterns of DNA by nucleases upon release from the nucleus, leading to distinct profiles of coverage with expressed genes having a dip in the coverage pattern over the TSS due to the lower nucleosome occupancy, as opposed to silent genes, which present a relatively flat coverage profile due to densely packed nucleosomes [17,18,19,20].

We applied a targeted sequencing approach, aiming for a higher depth of coverage than that of low-coverage WGS, targeting the TSS of 112 EBT-relevant genes. Importantly, distinct coverage profiles suggesting different nucleosome occupancy patterns across the studied regions were observed in 7/15 samples. The samples that were non-contributive for TSS analysis were those with lower CSF cfDNA concentrations.

Distinct patterns of nucleosome occupancy across the studied TSSs predicted genes to be over- or under-expressed in different tumor entities. However, clustering by disease type did not cluster all specific disease types together, with one MB clustering with the ATRT. This might be due to technical bias or poor cfDNA concentrations; a large difference in the cfDNA concentration between the two MB cases was observed (15 ng/mL vs. >40 ng/mL). In addition to nucleosome occupancy, many other molecular steps are involved in gene expression, and it has been suggested that altered promotor nucleosome positioning is an early event in gene silencing, with more permanent gene silencing linked to hypermethylation [41]. We show distinct TSS nucleosome footprints in different tumors; however, no robust tumor clustering could be achieved. Whereas expression profiles permit clustering according to tumor types, the absence of such clustering based on TSS nucleosome footprints might be due to the multiple molecular steps between TSS accessibility and steady state mRNA abundance. However, no direct correlation with expression data from paired samples could be studied. Future studies analyzing TSS nucleosome footprints across the whole genome, and correlating accessibility with expression profiles, could lead to the further comprehension of mechanisms implicated in gene regulation—in particular, in EBT.

## 5. Conclusions

Altogether, our study provides further insight into the clinical utility of cfDNA extracted from CSF obtained during clinical routine procedures, with informative results in 60% (15/25) of all cases and 83% (15/18) of cases studied by WES. Further studies, including correlation with methylation studies, will enable the further exploration of the observed specific nucleosome occupancy around the TSS of EBT-relevant genes. Future decision algorithms could propose CSF cfDNA-based molecular diagnosis as a first step if molecular diagnosis is compatible with neo-adjuvant chemotherapy prior to surgery. Similarly, it could be applicable in unresectable pediatric brain tumors, with a biopsy then proposed secondarily, if necessary. Finally, CSF cfDNA analyses could be integrated into disease surveillance during the treatment and follow-up.

## Figures and Tables

**Figure 1 cancers-15-03532-f001:**
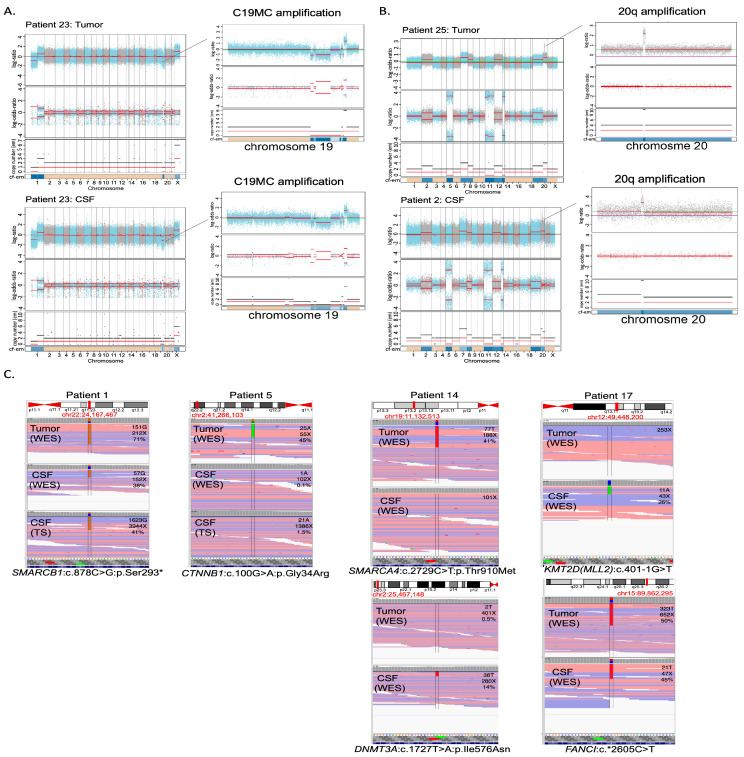
Genetic alterations detected in tumor tissue and cfDNA extracted from CSF. (**A**) Copy number profile in an EMTR showing amplification of C19MC in both samples (patient 23). (**B**) Copy number profile of an EBT NOS with an amplification on chr 20 in both samples, as well as gains of chromosomes 2, 7, 12 and 18 (patient 25). (**C**) SNVs detected in WES of the primary tumor, and WES and targeted sequencing of cfDNA, with labels indicating the number of reads supporting the SNVs, coverage at the given position and percentage of variant allele fraction: an *SMARCB1* mutation in ATRT (patient 1); a *CTNNB1* mutation seen by WES of the primary and targeted sequencing of cfDNA (patient 5). Heterogeneity of SNVS observed in an MB with an *SMARCA4* alteration seen in the primary tumor but not in CSF and *DNMT3A* seen in CSF but not in the primary tumor (patient 14). Heterogeneity of SNVs with *KMT2D* alterations seen in CSF but not in the primary tumor and an *FANCI* SNV seen in both (patient 17).

**Figure 2 cancers-15-03532-f002:**
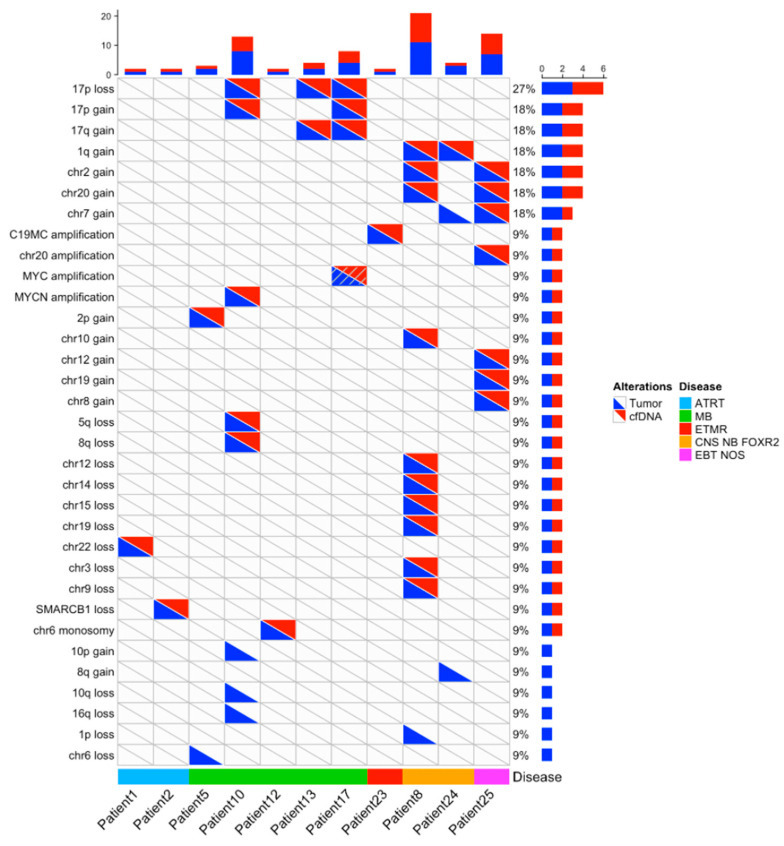
Copy number changes detected in cases with informative copy number analysis in both the tumor and CSF cfDNA. For 11 patients with an informative copy number analysis in CSF cfDNA (*x*-axis), copy number alterations are indicated, according to their detection in the tumor (blue triangle) or CSF cfDNA (red triangle) at diagnosis, or in one case at relapse (slashed red/blue triangle). Their frequency is indicated in bars on the right.

**Figure 3 cancers-15-03532-f003:**
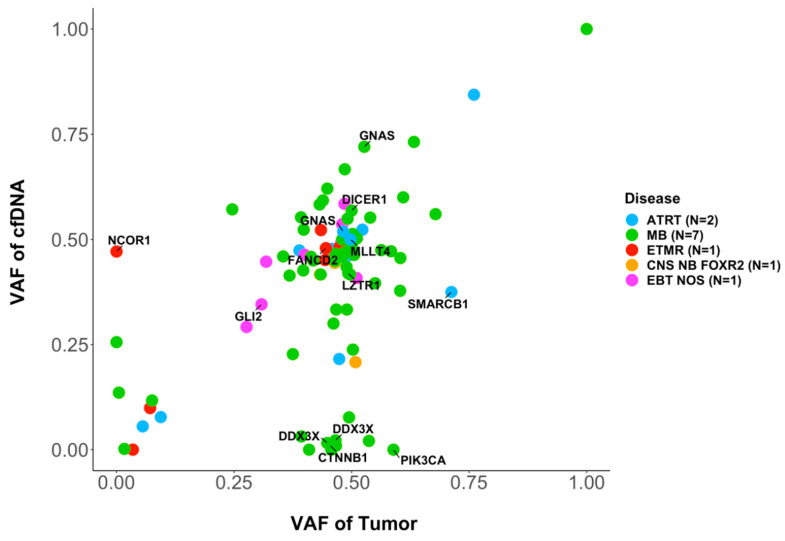
Variant allele fractions of SNVs of genes included in the targeted sequencing panel. In 15 cases with informative SNV analysis in the primary tumor and CSF cfDNA, the mutated allele fractions of SNVs in the tumor sample (*x*-axis) and in the CSF cfDNA (*y*-axis) are indicated. Pearson test showed a significant correlation (r = 0.51, *p* = 1.6 × 10^−7^).

**Figure 4 cancers-15-03532-f004:**
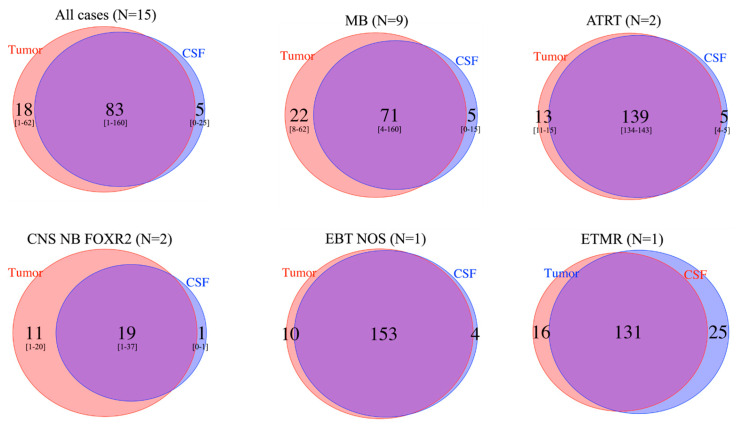
Comparison of number of SNVs from WES analysis in tumor and CSF cfDNA, among cases with informative analyses, followed by disease type. Among the 15 cases with informative SNV analysis in both the primary tumor and CSF cfDNA by WES, the total number of SNVs seen in the tumor (red circles), in the CSF cfDNA (blue circles) or in both (overlap) is indicated. Among the 15 patients, a mean of 83 SNVs (range 1–160) were observed in common between the tumor and CSF cfDNA, with 5 (range 0–25) and 18 (range 1–62) seen only in the CSF cfDNA and tumor, respectively, indicating clonal heterogeneity.

**Figure 5 cancers-15-03532-f005:**
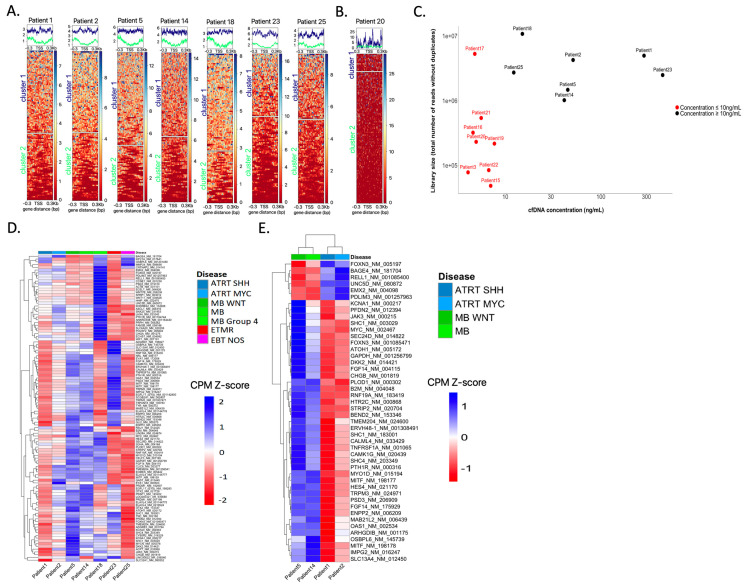
TSS nucleosome footprinting based on the coverage across the transcription start sites (TSS) of 112 genes in the TSS-targeted sequencing panel. (**A**) Coverage across the NDR regions surrounding the TSS distinguishes two different groups with specific patterns in seven cases using k-means clustering. Cluster 1 is highlighted in blue and cluster 2 ingreen. (**B**) Example of a case in which no distinct groups could be distinguished. (**C**) Library size (*y*-axis: total number of reads after removal of duplicates) depending on the CSF cfDNA concentration (*x*-axis); samples with un-informative TSS analyses are indicated in red. Pearson test showed no significant correlation (r = 0.17, *p* = 0.54). (**D**) Clustering on the total 112 NDR regions in seven samples for which the k-means clustering of genes into two groups was informative. Hierarchical clustering was performed on NDR regions only. The scores represent the counts per million (CPM) values of coverage sequencing after row scaling (row z-scores). The results of hierarchical clustering on the genes are represented by a dendrogram. Samples were ordered by disease type. The heatmap shows differences in coverage around the TSS across the seven samples. (**E**) Clustering on the 46 NDR regions selected after applying thresholds, on four samples (two ATRT and two MB cases). Hierarchical clustering was performed on NDR and on samples.

**Table 1 cancers-15-03532-t001:** Diagnostic molecular genetic alterations in 25 patients with embryonal brain tumors. Genetic alterations (SNVs and copy number alterations) detected in clinical molecular analysis are listed according to their detection in the primary tumor and in CSF cfDNA. Only SNVs of diagnostic impact are reported.

Patient Number	Primary Tumor Molecular Diagnosis: SNV	SNV in CSF cfDNA	Primary Tumor Molecular Diagnosis: CNA	CNA in CSF cfDNA
1	SMARCB1: c.851C>G/(p.Ser284*)	SMARCB1: c.851C>G/(p.Ser284*)	SMARCB1 loss	SMARCB1 loss
2			SMARCB1 loss	SMARCB1 loss
3	SMARCB1: c.601C>T/p.(Arg201*)	not found/CSF cfDNA non-contributive		
4			Gains: 2pter-p22.3 (subclone), 7, 17q12.2-qter Losses: 11pter-p11.12, 17q12.2-qter	not found/CSF cfDNA non-contributive
5	CTNNB1: Gly34Arg (c. 100G>A)	CTNNB1: Gly34Arg (c. 100G>A)	chr6 loss chr2pter-p13.2 gain	chr2pter-p13.2 gain
6			17q gain, 17p loss	not found/CSF cfDNA non-contributive
7			Losses: chr10, 2, 11, 13, 16, et 20	not found/CSF cfDNA non-contributive
8			Gains: 1q, 2, 10, 20 Losses: 1p, 3, 9, 12, 14, 15, 19, X	Gains: 1q, 2, 10, 20 Losses: 3, 9, 12, 14, 15, 19
9			17p loss, 17q gain	not found/CSF cfDNA non-contributive
10			Amplicon 2p24.3-2 containing MYCN. Losses: 5q31.2-ter 8q12.3-ter, 10q, 16q, 17pter-p11.2 Gains: 10p, 17p11.2-qter	Amplicon 2p24.3-2 containing MYCN. Losses: 5q31,2-ter, 8q12.3-ter, 17pter-p11.2 Gains: 17p11.2-qter
11	PTCH1: c.1359_1360insG (p.cys454valfs*43)	not found/CSF cfDNA non-contributive	9q loss 3q gain	not found/CSF cfDNA non-contributive
12	APC: c.3183_3187del/p.Gln1062* germline APC: c.3758del/p.Ser1253Leufs*12	APC: c.3183_3187del/p.Gln1062* germline APC: c.3758del/p.Ser1253Leufs*12	chr6 monosomy	chr6 monosomy
13			17p loss 17pq gain	17p loss 17pq gain
14				
15	CCND3 c.774_775delCTinsTG p.(Ser259Ala) COL3A1 c.946G>A p.(Ala316Thr) FANCD2 (c.1588C>T p.(Arg530*) NCKIPSD c.1650_1651delGCinsTT p.(Pro551Ser) PTPRC c.3452A>G p.(Lys1151Arg)	altered genes not included in targeted sequencing CSF cfDNA panel		
16			Gains: 1q(210.34 Mb-tel), 3p(tel-5.17 Mb)	not found/CSF cfDNA non-contributive
17			Losses: 17p(tel-18.91 Mb) harboring TP53 Gains: 17pq(19.14 Mb-tel)	Losses: 17p(tel-18.91 Mb) harboring TP53 Gains: 17pq(19.14 Mb-tel)
18			Losses: 11p-, 17p(tel-22.22 Mb) Gains: 4q(142.07 Mb-tel), 13q(90.81–96.80 Mb), 15q(54.53 Mb-tel), 17pq(25.28 Mb-tel) Amplicons: 7q(92.20–96.80 Mb) harboring CDK6	not found/CSF cfDNA non-contributive
19			Losses: 2q(213.21 Mb-tel), 8p(tel-10,18 Mb), 16q(57.83 Mb-tel), 17p(tel-18.15 Mb), X(78.64-Tel), X(16.68–17.87 Mb) Subclonal segmental losses: 4q(170.70 Mb-tel), 5q(170.87 Mb-tel), 8q(55.00–70.23 Mb), 9p(tel-11.78 Mb), 10q(107.71 Mb-tel), 12q(40.77–57.69 Mb)	No CN analysis for target seq data
20	CTNNB1: c.98C>G/p.(Ser33Cys) PIK3CA: c.311C>G/p.(Pro104Arg) (50, 3%; 193X)	CTNNB1: c.98C>G/p.(Ser33Cys)	chr6 monosomy	No CN analysis for target seq data
21			Losses: 17p(tel-18.91 Mb) Gains: 17pq(19.14 Mb-tel)	No CN analysis for target seq data
22			Losses: 16q(66.57 Mb-tel) Subclonal segmental losses: 8q(52.57 Mb-tel), 11q(75.97 Mb-tel), 13q(25.30–31.39 Mb), 13q(50.05–81.10 Mb) Gains: 7p(tel-4,88 Mb), 13q(20,28–25,27 Mb), 13q(31.45–49.99 Mb), 13q(81.14 Mb-tel), 17q(46.78 Mb-tel)	No CN analysis for target seq data
23			Amplification 19q13.41	Amplification 19q13.41
24	PTPRK p.(Thr395AspfsTer6)	not found/CSF cfDNA non-contributive	Gains: 1q, 7, 8q	Gains: 1q
25	KDR: Hist1144Asp TP53: Arg175His KRAS: Gly12Asp	not found/CSF cfDNA non-contributive	Gains: 2, 7, 12, 19. Amplification chr 20(30.5–30.8 Mb)	Gains: 2, 7, 12, 19. Amplification chr 20(30.5–30.8 Mb)

## Data Availability

Sequencing data (.vcf files) will be made available upon reasonable request.

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
