# Peer review of "Cell-Free DNA Extracted from CSF for the Molecular Diagnosis of Pediatric Embryonal Brain Tumors"

_cancers, 2023, doi:10.3390/cancers15133532_

Round 1
Reviewer 1 Report
In the current manuscript Schleiremacher et al. describes the usefulness of cfDNA isolated from CSF in pediatric CNS tumors. The manuscript is well written and the findings are clinically relevant. I would rather suggest two points
1-It could be interesting if the authors show the influence of the cfDNA input on the amount of mutations, CNV, SNV detected
2-Likely not to be included in the current manuscript but if cfDNA isolated from plasma is available try to detect mutations and other genomic alterations in this patients. During the surgery or shortly after it may be useful to obtain cfDNA from plasma and try to detect ctDNA that was released during the surgery
Author Response
Reviewer 1 :
1-It could be interesting if the authors show the influence of the cfDNA input on the amount of mutations, CNV, SNV detected.
Reply : We thank the reviewer for this important question. We now indicate the number of SNVs and CNAs detected in the CSF cfDNA depending on the CSF cfDNA concentration (supplementary figure 2). No correlation between the CSF cfDNA concentration and the total number of CNAs and SNVs could be observed.
p.7: No correlation between the cfDNA concentration and the total number of CNAs and SNVs was observed (Supplementary figure 2).
2-Likely not to be included in the current manuscript but if cfDNA isolated from plasma is available try to detect mutations andother genomic alterations in these patients. During the surgery or shortly after it may be useful to obtain cfDNA from plasma and try to detect ctDNA that was released during the surgery
Reply : We agree that it would be of interest to study cfDNA extractted from blood plasma for these patients. However unfotunately no blood plasma was available for these patients . We have adressed this in the discussion :
- 14 : Further studies might also explore the possibility of analysis of cfDNA extracted from blood in pediatric patients with EBT.

Reviewer 2 Report
The authors of the presented manuscript investigated liquids of pediatric brain tumor patients and established a custom sequencing panel for analysis them. They demonstrated an alteration accordance of 60% (SNVs or CNAs) between the sequenced tissues and their matching CSF samples. However, some information is still missing and need to revised and changed. Especially the overall structure of the manuscript is not clear, reading and re-writing would be essential to improve the paper. There are many English-language mistakes – please carefully read the paper again and include better punctuation.
Introduction:
The last block of the introduction has a bigger letter size then the remaining manuscript.
Methods:
In section 2.2 the authors write 5-6 droplets and correspond this volume to >300µl and in the section 2.4. the same droplet number correspond to 500µl of CSF aliquots. I recommend to use only µl as general unit – droplets are not real units- their volume per drop is variable (depending on the used pipet size!). The authors can write a minimum of 300µl of CSF was used (for example).
2.3. Please change the word order for the patients: …., was performed in patients 6 and 18, respectively.
2.5. Change the number to 112 genes
Results:
The first paragraph should be included in the methods section.
The first figure was not readable – therefore I can’t review this section – please provide a better resolution of figure 1.
Section 3.2. What did the authors mean by the following sentence:” These approaches identified large-scale CNA in tumor DNA in all but one case, …”? Should it mean in all except one case?
The fraction of ctDNA is mentioned – it would be of impact, if the authors provide a graph for this result.
Section 3.3. Is there any reason or explanation why case 3 has low cfDNA concentration? Was the volume of CSF too low due to the age of the patient, or any other reasons?
In table 1 the authors write for patient 9 a 17p loss and 17p gain, is this a typing mistake? Also in figure 2, patient 17 has both CNAs (17p loss and 17p gain), where this different time points or are these segmental losses or gains of the chromosome? Can the authors please give a short comment on these two cases? Next, for patient 25 the CNA results are missing in the table, but are included in figure 2. In addition, please give more detailed information on the figure legend.
Section 3.5. the authors mentioned “In 15 cases with informative SNV analysis” in table 1 only 4 additional cases showed informative SNVs. Can the authors comment on these two different numbers of cases?
3.6. “Taking into account all SNVs observed among the 15 cases with informative SNV analysis in the primary tumor and CSF cfDNA by WES, a mean of 83 (range 1- 60) SNVs were observed in common between the tumor and CSF cfDNA, with 5 (range 0-25) and 18 (range 1-62) seen only in the CSF cfDNA and tumor, respectively, indicating clonal heterogeneity in all EBT types (Figure 4).” Please elaborate this confusing long sentence.
“Comparison of number of SNVs from WES analysis in tumor and CSF cfDNA, among cases with informative analyses, followed by disease type.” How is informative analysis defined? – not given in table 1
3.7. Please add representative case in the main text for figure 5B.
Discussion:
Cell-free DNA should be announced in the introduction or when it appears at first time in the text (not in the discussion).
“Furthermore, we show that this cfDNA, although obtained in small quantities, can lead to informative results for CNA and SNV calling in 11/18 and 15/25 cases, respectively.” Can the authors comment on how they were summarized? In the results part 11/25 or 15/25 cases were pronounced.
Please indicate what coverage you got by WES of CSF depending on the respective DNA amount used.
“All cases with a ctDNA content > 20% showed copy number changes in the CSF cfDNA”. Which cases showed a higher ctDNA content? It would be great, if the authors can include this information in the table and provide a graph with the measurements of the bioanalyzer (as mentioned for 3.2).
“Second, it will be feasible to use these approaches for the assessment of tumor burden, and the evaluation of minimal residual disease [33],[37]”. Can the authors comment and discuss the low number of informative results (only 60%)? Is this due to the limited cases? Maybe a ROC curve would also be a nice tool to show the sensitivity and specificity of each of the described techniques?
“We show distinct TSS nucleosome footprints in different tumors; however, the absence of robust tumor clustering as opposed to that seen by expression profiles could be linked an absence of a direct correlation between TSS nucleosome footprints and expression levels in the studied genes.” A word seems to be missing in this sentence, it does not make much sense this way.
Supplementary figures:
In supplementary figure legend 1 there is a partition in A, B and C, but the B is missing next to the graph.
Supplementary table 1: The authors should create two individual tables, one for the clinical and one for the biological information. Otherwise, it is overwhelming. Further, in the second part of table 1 the first row is unreadable; the hyphenations of the words are done without following any grammar rules. Please change the complete table and describe all abbreviations in the legend.
Supplementary table 1: negative not negatif
What are idays
In supplementary figure 2 Resolution of Chr8 CGHarray is not high enough to check for MYC amplification – please provide higher resolution.
Is it possible to perform WES on the primary tumor to check whether it has been there already as determined by another method? Or FISH?
“At relapse, WES on the primary tumor reveals MYC amplification” WES of the primary tumor? Was that a retrospective analysis of the primary tumor or is it an analysis of the recurrent tumor tissue?
They write NEC not elsewhere classified and NOS not otherwise specified – please stick to one abbreviation
Is the relapse yes/no in the table the timepoint of sampling or is it a general information on the status of the patient at last follow up? – according to the text there was just one CNS positive patient – highlight in Supplementary table 1.
The overall structure of the manuscript is not clear, reading and re-writing would be essential to improve the paper. There are many English-language mistakes – please carefully read the paper again and include better punctuation.
Author Response
Reviewer 2 :
There are many English-language mistakes – please carefully read the paper again and include better punctuation.
Reply : We apologize for any mistakes. The manscuript has been read and corrected by two native English speakers, and the punctuation has been revised. Furthermore we have restructured the results section for further clarity, as suggested by the reviewer.
- Introduction:
The last block of the introduction has a bigger letter size then the remaining manuscript.
Reply : we have corrected the letter size of the last paragraph of the introduction.
2.Methods:
In section 2.2 the authors write 5-6 droplets and correspond this volume to >300μl and in the section 2.4. the same droplet number corresponds to 500μl of CSF aliquots. I recommend to use only μl as general unit – droplets are not real units- their volume per drop is variable (depending on the used pipet size!).The authors can write a minimum of 300μl of CSF was used (for example).
Reply : we have indicated that a minimum of 300 µl of CSF was used for extraction of cfDNA .
- 4 : For cfDNA studies, a minimum of 300 µl of CSF was prepared…
- 4 : cfDNA was extracted from a minimum of 300 µl of CSF…
2.3. Please change the word order for the patients: …., was performed in patients 6 and 18, respectively.
Reply : In this sentence we indicate the number of patients for whom analysis by WES on tumor and germline material could be performed. We have modified the sentence for further clarification
- 4 : Whole-exome sequencing (WES) of genomic DNA extracted from primary tumors, and paired germline genomic DNA, was performed in a total of 18 and 6 patients, respectively (Supplementary methods)
2.5. Change the number to 112 genes
Reply : we have modified the sentence as requested :
- 4 : 112 genes which contribute to the genetic classification of embryonic tumors…
- Results:
The first paragraph should be included in the methods section.
Reply : As this paragraph refers to results obtained following cfDNA extraction, and the correlation of the findings with clinical parameters, with further reference to supplementary table 1, we would like to propose to leave this paragraph in the « Results » section. We hope that the reviewer can accept our proposition.
The first figure was not readable – therefore I can’t review this section – please provide a better resolution of figure 1.
Reply : A better resolution of figure 1 has been provided and inserted in the Word file.
Section 3.2. What did the authors mean by the following sentence:” These approaches identified large-scale CNA in tumor DNA in all but one case, …”? Should it mean in all except one case?
Reply : We confirm that we mean «… in all except one case… ». We have modified the sentence accordingly.
- 7: These approaches identified large-scale CNAs in tumor DNA in all cases except one…
The fraction of ctDNA is mentioned – it would be of impact if the authors provide a graph for this result.
Reply : The ctDNA fraction in cfDNA is indicated in supplementary table 1. This information has beed further detailed in the text as follos :
- 7 : The fraction of ctDNA in the CSF cfDNA was determined in samples sequenced by WES, with a range of 0.15 – 0.86 (Supplementary table 1b). The highest ctDNA content was observed in the ETMR case.
Section 3.3. Is there any reason or explanation why case 3 has low cfDNA concentration? Was the volume of CSF too low due to the age of the patient, or any other reasons?
Reply : Low cfDNA concentrations were observed for patient 3, and in several other patients (patient 16, patient 17 ; supplementary table 1). No explanation was found to explain the variable cfDNA concentrations, and no correlation with age was observed.
- 5: No correlation between cfDNA concentration and age, disease status (localized versus metastatic disease), the presence/absence of a postsurgical residue, or the delay between surgery and CSF sampling was observed (supplementary figure 1).
In table 1 the authors write for patient 9 a 17p loss and 17p gain,is this a typing mistake?
Reply : For patient 9 indeed a typing error occured in table 1. For this patient a 17p loss and 17q gain was oberved. We have corrected this in table 1.
Also in figure 2, patient 17 has both CNAs (17p loss and 17p gain), where this these different time points or are these segmental losses or gains of the chromosome?
Reply : Indeed for this patient with a breakpoint on chromosome 17p, a segmental 17p loss followed by a segmental 17p gain is observed, as indicated also in table 2. The copy number profile is shown in supplementary figure 4. In patient 10, a segmental 17p loss followed by a segmental 17p gain is also observed. This is a recurrent finding in medulloblastoma. We have indicated this in the text.
- 7: …with amplifications of MYCN, a region on chromosome 20 and a region on chromosome 19 (n=4), homozygous deletions, or heterozygous copy number losses or gains, including loss and gain of chromosome 17p
Next, for patient 25 the CNA results are missing in the table, but are included in figure 2.
Reply : we have included the CNA results for patient 25 in table 1.
Table 1 : Gains : 2, 7, 12, 19. Amplification chr 20(30.5-30.8Mb)
In addition, please give more detailed information on the figure legend.
Reply : Legend to figure 2 has been detailed
- 8 : Copy number changes detected in cases with informative copy number analysis both in tumor and CSF cfDNA. For 11 patients with an informative copy number analysis in CSF cfDNA (x-axis), copy number alterations are indicated, according to their detection in tumor (blue triangle) or CSF cfDNA (red triangle). Their frequency is indicated in bars on the right.
Section 3.5. the authors mentioned “In 15 cases with informative SNV analysis” in table 1 only 4 additional cases showed informative SNVs. Can the authors comment on these two different numbers of cases?
Reply : In table 1 we have reported the SNVs with clinical diagnostic relevance or impact. Not all SNVs are listed. The total number of SNV sis listed in supplementary table 1. We have clarified this in the legend to table 1
- 10 : Table 1. Diagnostic molecular genetic alterations in 25 patients with embryonal brain tumors. Genetic alterations (SNVs and copy number alterations) detected in clinical molecular analysis are listed according to their detection in the primary tumor, and in CSF cfDNA. Only SNVs of diagnostic impact are reported.
3.6. “Taking into account all SNVs observed among the 15 caseswith informative SNV analysis in the primary tumor and CSFcfDNA by WES, a mean of 83 (range 1- 60) SNVs wereobserved in common between the tumor and CSF cfDNA, with 5(range 0-25) and 18 (range 1-62) seen only in the CSF cfDNAand tumor, respectively, indicating clonal heterogeneity in allEBT types (Figure 4).” Please elaborate this confusing long sentence.
Reply : we have modified this sentence as suggested.
Page 12 : Taking into account all SNVs observed among the 15 cases with informative SNV analysis in the primary tumor and CSF cfDNA by WES, a mean of 83 (range 1-60) SNVs were observed in common between the tumor and CSF cfDNA. Furthermore, a mean of 5 SNVs (range 0-25) and 18 SNVs (range 1-62) were seen only in the CSF cfDNA or only in the tumor, respectively, indicating clonal heterogeneity in all EBT types (Figure 4).
“Comparison of number of SNVs from WES analysis in tumorand CSF cfDNA, among cases with informative analyses,followed by disease type.” How is informative analysis defined?– not given in table 1
Reply : The definition of an informative analysis, for SNV and CNA analysis, is indicated in paragraph 3.2.
- 7 : For CSF cfDNA analysis, sequencing results were considered informative in the case of identification of at least one CNA or SNV. cfDNA extracted from CSF was sequenced by WES alone in 10 cases, the TSS targeted panel alone in seven cases, or both approaches in eight cases. These approaches identified CNAs and SNVs in 11 cases, and SNVs only in 4 additional cases (Figure 1, supplementary Table 1b).
3.7. Please add representative case in the main text for figure 5B.
Reply : Figure 5b is referenced in the main text.
- 14 : In 8 other samples, these two distinct patterns could not be clearly distinguished (Figure 5B).
- Discussion:
Cell-free DNA should be announced in the introduction or when it appears at first time in the text (not in the discussion).
Reply : The abbreviation for cell-free DNA (cfDNA) has been defined in the introduction. We have removed the erronous full text at the beginning of the discussion.
- 3 : The study of cell-free DNA (cfDNA), small DNA fragments released into either the blood stream or other body fluids
- 14 : Liquid biopsies are revolutionary tools to detect tumor-specific genetic alterations in body fluids, and cfDNA…
“Furthermore, we show that this cfDNA, although obtained in small quantities, can lead to informative results for CNA and SNV calling in 11/18 and 15/25 cases, respectively.” Can the authors comment on how they were summarized? In the results part 11/25 or 15/25 cases were pronounced.
Reply : We apologize for the confusion with regards to the reporting of informative results, which depended on whether we considered all cases, or only cases for which WES analyses were performed. We have clarified this in the abstract, results and discussion session.
p.2 : Conclusion: CSF cfDNA sequencing yielded informative results in 60% (15/25) of all cases, with results informative in 83% (15/18) of all cases analysed by WES.
p.14 : We confirm in our series that CSF can be a reliable source of cfDNA in pediatric EBT, although obtained in small quantities. Furthermore, we show that this cfDNA, used for WES in 18/25 cases, leads to informative results for CNA and SNV identification in 11/25 and 15/25 cases, respectively.
- 16 : …with informative results in 60% (15/25) of all cases, and 83% (15/18) of cases studied by WES.
Please indicate what coverage you got by WES of CSFdepending on the respective DNA amount used.
Reply : The overall coverage is indicated in supplementary table 2. We have furthermore indicated the coverage according to the cfDNA concentration in supplementary figure 2.
“All cases with a ctDNA content > 20% showed copy number changes in the CSF cfDNA”. Which cases showed a higher ctDNA content? It would be great, if the authors can include this information in the table and provide a graph with the measurements of the bioanalyzer (as mentioned for 3.2).
Reply : The ctDNA content is indicated in supplementary table 1b. No correlation between ctDNA content and the number of identified CNAs or SNVs could be observed. This has been indicated in the text.
- No correlation between the cfDNA concentration and the total number of CNAs and SNVs was observed (Supplementary figure 2).
“Second, it will be feasible to use these approaches for the assessment of tumor burden, and the evaluation of minimalresidual disease [33,37]”. Can the authors comment and discuss the low number of informative results (only 60%)? Is this due to the limited cases? Maybe a ROC curve would also be a nice tool to show the sensitivity and specificity of each of the described techniques?
Reply : The low number of informative results might be due to the limited number of cases, the very low CSF volume and the retrospective sample collection, with samples obtained after surgical resection in a majority of cases. We have discussed this in more detail in the discussion section :
- 14 Non-informative cases might be linked to the small amounts of CSF, low ctDNA content, or the delay until CSF collection after surgery, particularly for cases with complete excision (mean of 17.9 days after surgery; range -3 days to +67 days). In prospective studies, CSF collection at diagnosis, either before or during surgery, might increase the utility of cfDNA analysis.
“We show distinct TSS nucleosome footprints in different tumors;however, the absence of robust tumor clustering as opposed to that seen by expression profiles could be linked an absence of a direct correlation between TSS nucleosome footprints and expression levels in the studied genes.” A word seems to be missing in this sentence, it does not make much sense this way.
Reply : we have modifed the sentence for further clarification.
- 15 : We show distinct TSS nucleosome footprints in different tumors; however no robust tumor clustering could be achieved. Whereas expression profiles permit clustering according to tumor types, the absence of such clustering based on TSS nucleosome footprints might be due to the multiple molecular steps between TSS accessibility and steady state mRNA abundance.
Supplementary figures:
In supplementary figure legend 1 there is a partition in A, B and C, but the B is missing next to the graph.
Reply : The B has been added to the graph.
Supplementary table 1: The authors should create two individual tables, one for the clinical and one for the biological information. Otherwise, it is overwhelming. Further, in the second part of table 1 the first row is unreadable; the hyphenations of the words are done without following any grammar rules. Please change the complete table and describe all abbreviations in the legend.
Supplementary table 1: negative not negatif
What are idays
Reply : We have corrected the supplementary table 1 according to the reviewer’s request, correcting the hyphenation and spelling. The tablme has been separated in supplementary table 1A and Supplementary Table 1B.
In supplementary figure 2 Resolution of Chr8 CGHarray is not high enough to check for MYC amplification – please provide higher resolution.
Is it possible to perform WES on the primary tumor to check whether it has been there already as determined by another method? Or FISH?
“At relapse, WES on the primary tumor reveals MYC amplification” WES of the primary tumor? Was that a retrospective analysis of the primary tumor or is it an analysis of the recurrent tumor tissue?
Reply : For this patient, at diagnosis, aCGH was performed. No additional analyses could be performed on the sample obtained at diagnosis. At relapse, WES was performed on the recurrent tumor. We have clarified this in the text. We have increased the resolution and zoom of the different analyses.
- 23 : At relapse, WES on the recurrent tumor reveals MYC amplification.
They write NEC not elsewhere classified and NOS not otherwisespecified – please stick to one abbreviation
Reply : this has been rectified ; the abbreviation NOS is used throughout the text.
Is the relapse yes/no in the table the timepoint of sampling or is it a general information on the status of the patient at last followup?
Reply : The colum « relapse » refers to the patient follow-up ; the timepoint of sampling is indicated in a separate colum in supplementary table 1.
According to the text there was just one CNS positive patient – highlight in Supplementary table 1.
Reply : patients with CSF cytologically positive examinations are indicated in the supplementary table 1a.

Reviewer 3 Report
Comments
This is a well-written manuscript titled “Cell-free DNA extracted from CSF for molecular diagnosis of pediatric embryonal brain tumors” demonstrated that liquid biopsies, cell-free DNA extracted from CSF can be used for whole exome sequencing (WES), with informative results in 83% for pediatric embryonal brain tumors. The further author revealed that CSF cfDNA sequencing yielded informative results in 60% of cases (15/25), with WES feasible in 83% (15/18). This work is of great importance where getting tumor biopsy is limited and liquid biopsies will be the revolutionary tools to detect tumor-specific genetic alterations in body fluids, and cell-free DNA (cfDNA) can be used for molecular diagnosis in cancer patients. This manuscript will be of good interest to the scientific community. The manuscript can be accepted in its present form with minor revisions.
1. It will be interesting to map alteration in histone modification in CSF from pediatric embryonal brain tumors.
The manuscript is well-written and easy to understand. The manuscript can be accepted with minor proofreading and spelling check.
Author Response
Reviewer 3 :
Reviewer 3 indicates that it will be interesting to map alteration in histone modification in CSF from pediatric embryonal brain tumors.
Reply : We fully agree with this observation. Unfortunately it is beyond the scope of this manuscript to analyse histone modifications in CSF from pediatric EBT. We have added this to the discussion
- 16 : It might also be of interest to determine whether histone modifications could be determined directly in the CSF of pediatric EBT patients.
